# Deep Learning-Based Medical Ultrasound Image and Video Segmentation Methods: Overview, Frontiers, and Challenges

**DOI:** 10.3390/s25082361

**Published:** 2025-04-08

**Authors:** Xiaolong Xiao, Jianfeng Zhang, Yuan Shao, Jialong Liu, Kaibing Shi, Chunlei He, Dexing Kong

**Affiliations:** 1College of Mathematical Medicine, Zhejiang Normal University, Jinhua 321004, China; 1359152602@zjnu.edu.cn (X.X.); shaoy0625@zjnu.edu.cn (Y.S.); liujialong@zjnu.edu.cn (J.L.); shikaibing@zjnu.edu.cn (K.S.); chunlei@zjnu.edu.cn (C.H.); dxkong@zju.edu.cn (D.K.); 2School of Computer Science and Technology (School of Artificial Intelligence), Zhejiang Normal University, Jinhua 321004, China; 3Puyang Institute of Big Data and Artificial Intelligence, Puyang 457006, China; 4School of Mathematical Sciences, Zhejiang University, Hangzhou 310027, China

**Keywords:** deep learning, ultrasound image, datasets, segmentation, ultrasound video, review, segment anything model

## Abstract

The intricate imaging structures, artifacts, and noise present in ultrasound images and videos pose significant challenges for accurate segmentation. Deep learning has recently emerged as a prominent field, playing a crucial role in medical image processing. This paper reviews ultrasound image and video segmentation methods based on deep learning techniques, summarizing the latest developments in this field, such as diffusion and segment anything models as well as classical methods. These methods are classified into four main categories based on the characteristics of the segmentation methods. Each category is outlined and evaluated in the corresponding section. We provide a comprehensive overview of deep learning-based ultrasound image segmentation methods, evaluation metrics, and common ultrasound datasets, hoping to explain the advantages and disadvantages of each method, summarize its achievements, and discuss challenges and future trends.

## 1. Introduction

Continuous improvements in artificial intelligence, especially in deep learning techniques, are helping to identify, classify, and quantify patterns in clinical images [1]. Applying deep learning to lesion detection, classification, and other related tasks, deep learning demonstrates superior performance compared to traditional techniques and even surpasses some highly skilled medical professionals in certain tasks [2]. Deep learning is a type of machine learning that enables computers to learn from experience and comprehend the environment through a hierarchical conceptual framework [3]. Models of deep learning have the capability to learn and extract intricate patterns and features from vast datasets, making them exceptionally proficient in areas such as image recognition, natural language processing, medical diagnosis, and autonomous vehicles. Applying some deep learning models to medical images can provide very critical diagnostic evidence for clinical diagnosis. Deep learning has revolutionized medical image analysis through its applications in core computer vision tasks, including image classification, lesion detection [4], semantic segmentation, multimodal registration, reconstruction from sparse data, and cross-modality synthesis. Among these, the foundational triad—classification, detection, and segmentation—continues to dominate clinical implementations [5], while emerging tasks like deformable registration and pathology quantification are increasingly addressing complex diagnostic challenges. Medical ultrasound imaging is a safe, painless, and non-invasive diagnostic tool that has become well-established in clinical practice. This is primarily due to its advantages of low cost, portability, and widespread availability compared to other medical imaging modalities [6]. The segmentation of medical images plays a crucial role in clinical diagnosis, and accurate segmentation can facilitate more precise quantitative and qualitative analysis. It plays a crucial role in the diagnosis of various medical conditions, including internal organ tumors, and has also become a standard method for prenatal diagnosis [7]. Image segmentation involves the extraction of regions of interest from the background in an ultrasound image, which significantly influences the quality of the final analysis [8]. However, due to the complex imaging structure of ultrasound images, a variety of artifacts and noises, including high scattering noise due to the coherence of ultrasound imaging [9,10], low signal-to-noise ratios and intensity inhomogeneities, and the need for skilled operators, pose additional challenges to image segmentation. Figure 1 presents a series of representative ultrasound images obtained from a typical breast ultrasound examination.

A variety of methodologies have been proposed to address the aforementioned challenges. Some of the commonly used methods include the application of active contour models [11,12] for image and video segmentation based on levels set by spectral clustering and normalization criterion (NCut) [13], segmentation methods based on texture and shape prior [14], statistical shape models using Gabor filters [15], using a radial basis function neural network for classification to differentiate potential segmented regions [16], model initialization and efficient DDC (distance dynamic contour) contour refinement [9], the design of partial differential equation-based streams for maximum likelihood segmentation in scene targets [17], and a priori property profiles for fully automatic segmentation [18]. Despite extensive research into ultrasound image and video segmentation, it continues to be a prominent area of focus in current studies. In recent years, various deep learning-based ultrasound image and video segmentation methods have been proposed, aiming to leverage the promising potential of artificial intelligence to assist doctors in addressing the challenges posed by complex imaging structures, various artifacts, and noise in medical ultrasound image segmentation, thereby improving segmentation accuracy [19]. Therefore, this paper classifies deep learning approaches in ultrasound image segmentation based on the neural network structure as well as in combination with other recently emerged methods, including: U-Net neural networks and its variants, fully convolutional neural networks (FCNs), recurrent neural networks (RNNs), generative adversarial networks (GANs), deep reinforcement learning (DRL) methods, diffusion models, weakly supervised learning (WSL), and segment anything models (SAMs). Furthermore, this paper elaborates on several evaluation methods for image segmentation and introduces a number of high-quality, publicly available datasets. Figure 2 presents statistical data regarding the classification of the previously mentioned deep learning models.

**Figure 1 sensors-25-02361-f001:**
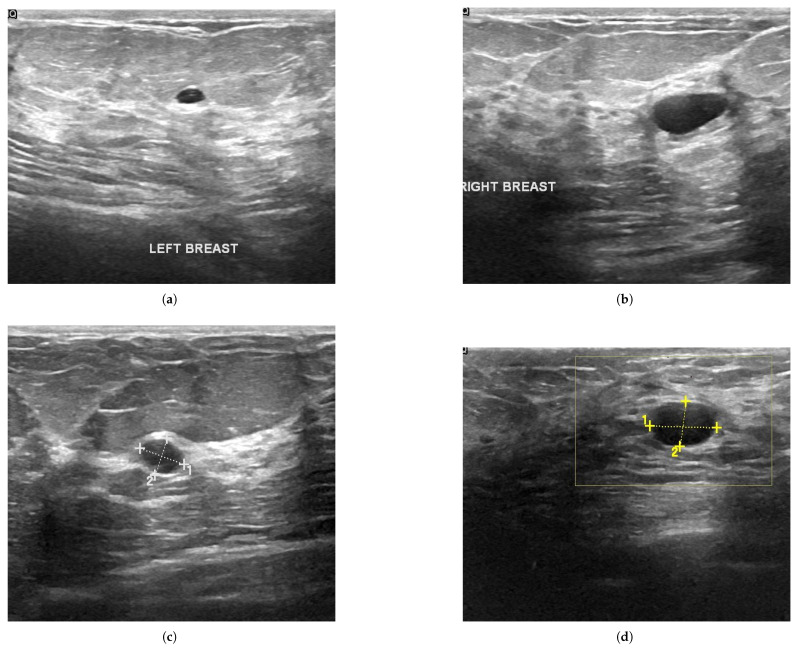
Ultrasound images of breast. The ultrasound images of the left and right breasts, denoted as (**a**) and (**b**), respectively, show relatively low contrast. Images (**c**,**d**) have annotations on the areas of the breast nodules. Ultrasound image showing a lesion marked with measurement calipers. The crosses indicate the boundaries used to calculate the transverse (1) and longitudinal (2) diameters. The rectangular region highlights the area of interest for further analysis. https://datasetninja.com/breast-ultasound-images (accessed on 3 January 2025).

The main contributions of this paper are listed as follows:To our knowledge, this is the first systematic review of deep learning-based medical ultrasound image and video segmentation methods, including common evaluation methods and datasets.With a survey of more than 80 papers, we provide comprehensive introductions, analyses, and detailed statistics on recent and classical publications from different perspectives (such as methods and evaluation criteria) in the related sections, tables, and figures.According to our survey and statistics, we reasonably put forward challenges and future trends in the Discussion and Conclusion section.

**Figure 2 sensors-25-02361-f002:**
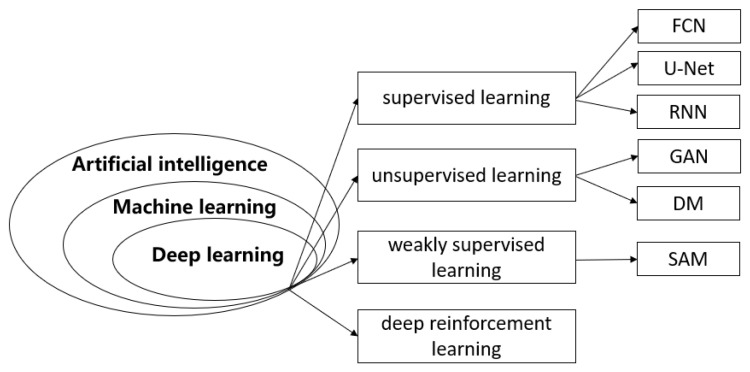
Deep learning is a branch of artificial intelligence and machine learning. According to the classification of the learning paradigm, it is classed as more detailed, secondary classification learning.

The remainder of our paper is organized as follows: Section 2 introduces common architectures in deep learning and some recently emerging artificial intelligence models combined with deep learning. In Section 3, we discuss related work on deep learning-based medical ultrasound image and video segmentation methods and review publicly available medical ultrasound image and video datasets. In addition, for some deep learning methods with more data, the literature is summarized in a table, including the ROI, dataset, corresponding model, and so on. Section 4 presents common evaluation methods. In the Section 5 we draw conclusions by discussing issues related to deep learning methods for medical ultrasound image and video segmentation and speculate on future technical developments and trends in this research field.

## 2. Description of Deep Learning Methods Based on Learning Paradigm Classification

In recent years, deep learning has become the dominant paradigm in medical ultrasound image and video segmentation. To better capture the diversity of approaches and address concerns about classification clarity, we categorize existing methods based on learning paradigms, namely, supervised learning, semi-supervised learning, weakly supervised learning, unsupervised learning, deep reinforcement learning, diffusion models, and the segment anything model (SAM), which are advanced techniques applied in medical ultrasound image segmentation. Although SAM is pre-trained in a fully supervised manner on a large-scale dataset, it performs segmentation during inference using only a small amount of weak supervision (prompts). Therefore, it is categorized under the weakly supervised learning paradigm. Each paradigm reflects a different degree of reliance on annotated data and offers unique advantages and challenges.

### 2.1. Supervised Learning

Supervised learning methods require fully annotated data (i.e., pixel-level labels for each image) to train segmentation models. These methods typically offer the highest accuracy due to the richness of supervision signals. Prominent approaches are described below.

#### 2.1.1. Fully Convolutional Neural Networks

FCNs are a type of deep learning model designed for semantic image segmentation tasks. A key feature of FCNs compared to traditional CNNs is their ability to accept input images of arbitrary size and output dense predictive maps of the corresponding size, which makes FCNs particularly suitable for image segmentation tasks. Long et al. proposed fully convolutional neural networks in 2015 [20], which opened a new chapter in the use of deep learning for image segmentation. FCNs adapt AlexNet, VGG networks, and GoogLeNet to full convolutional networks based on the end-to-end, pixel-to-pixel training of convolutional networks, starting from supervised pretraining without the need for complex processing in state-of-the-art semantic segmentation [21]. Global convolutional networks transfer their learned representations to the segmentation task by fine-tuning, defining a skip architecture used for more accurate segmentation. Since learning and inference are performed through intensive feed-forward computation and backpropagation, the in-network upsampling layer facilitates pixel-level prediction and learning in networks that incorporate subsampling pools. Figure 3 shows a typical FCN network architecture.

#### 2.1.2. U-Net Convolutional Neural Networks

U-Net is a variant convolutional network model based on the traditional CNN convolutional network model, which is still one of the most commonly used network architectures due to its excellent performance since it was proposed by Ronneberger et al. in 2015 [22]. An early popular approach to deep learning for semantic segmentation was patch classification, which has limitations due to its requirement for fixed-size input images and relatively slow processing speed; at the same time, it generated large amounts of redundant data [23]. U-Net modifies and extends the traditional convolutional network by incorporating a symmetric contraction path for robust context capture, a symmetric expansion path for precise localization, and enhanced upsampling layers that utilize multiple feature channels to transmit contextual information to higher-resolution layers. This architecture enables more accurate segmentation with a reduced number of training images. U-Net improves resolution by adding a continuous layer to the traditional shrinking grid and replacing the pooling layer with an upsampling layer. The network does not have any fully connected layers and uses only the effective portion of each convolution, i.e., the segmentation map contains only the pixels available in the full context of the input image. Extensive data augmentation through the application of elastic deformations to the available training datasets, coupled with the utilization of a weighted loss function to distinguish objects in contact within the same class, renders the network well-suited for a diverse array of biomedical challenges. Figure 4 shows a generic U-Net architecture. Due to the simplicity and superior performance of the U-Net architecture, various U-Net-like methods continue to emerge, such as Res-UNet [24] and Dense-UNet [25], and it has been extended to other medical image analysis tasks, such as image denoising [26], image registration [27], and more.

#### 2.1.3. Recurrent Neural Networks

RNNs are commonly used in natural language processing (NLP). In contrast to feed-forward networks, recurrent neural networks are suitable for processing temporal signals [28]. A typical RNN consists of three parts: sequential input data, a hidden state, and sequential output data. The RNN utilizes sequential information and executes the same operation for each element of the sequence, where the output is contingent upon prior computations. The RNN is capable of processing sequences of arbitrary length, and the internal state is used to model and “remember” the previously processed information. Long short term memory (LSTM) and gated recurrent units (GRUs) are two of the most prevalent architectures in recurrent neural network models, capable of effectively simulating long-term dependencies [29].

LSTM networks manage information flow through three gates—forget gate, input gate, and output gate—and a cell state. These components enable precise control over what information is retained, discarded, or updated, effectively solving the long-term dependency problem in traditional RNNs. The internal operations of an LSTM unit are described by the following equations:(1)$$\begin{array}{ccc} f_{t} & =\sigma (W_{f}\cdot \left[h_{t-1},x_{t}\right]+b_{f}),\; & //\;\mathrm{Forget}\;\mathrm{gate} \end{array}$$(2)$$\begin{array}{ccc} i_{t} & =\sigma (W_{i}\cdot \left[h_{t-1},x_{t}\right]+b_{i}),\; & //\;\mathrm{Input}\;\mathrm{gate} \end{array}$$(3)$$\begin{array}{ccc} {\tilde{C}}_{t} & =\tanh(W_{C}\cdot \left[h_{t-1},x_{t}\right]+b_{C}),\; & //\;\mathrm{Candidate}\;\mathrm{cell}\;\mathrm{state} \end{array}$$(4)$$\begin{array}{ccc} C_{t} & =f_{t}⊙C_{t-1}+i_{t}⊙{\tilde{C}}_{t},\; & //\;\mathrm{Cell}\;\mathrm{state}\;\mathrm{update} \end{array}$$(5)$$\begin{array}{ccc} o_{t} & =\sigma (W_{o}\cdot \left[h_{t-1},x_{t}\right]+b_{o}),\; & //\;\mathrm{Output}\;\mathrm{gate} \end{array}$$(6)$$\begin{array}{ccc} h_{t} & =o_{t}⊙\tanh\left(C_{t}\right),\; & //\;\mathrm{Hidden}\;\mathrm{state}\;\mathrm{output} \end{array}$$
where σ denotes the sigmoid activation function, tanh is the hyperbolic tangent function, and ⊙ represents element-wise multiplication. The gates determine how much of the previous hidden state $h_{t-1}$ and current input $x_{t}$ should influence the current step.

A GRU is a simplified variant of LSTM that merges the cell state and hidden state into a single hidden representation. GRU uses two gates—update gate and reset gate—to control the flow of information, reducing the number of parameters while preserving sequence modeling capabilities. The internal operations of a GRU unit are defined as follows:(7)$$\begin{array}{ccc} r_{t} & =\sigma (W_{r}\cdot \left[h_{t-1},x_{t}\right]+b_{r}),\; & //\;\mathrm{Reset}\;\mathrm{gate} \end{array}$$(8)$$\begin{array}{ccc} z_{t} & =\sigma (W_{z}\cdot \left[h_{t-1},x_{t}\right]+b_{z}),\; & //\;\mathrm{Update}\;\mathrm{gate} \end{array}$$(9)$$\begin{array}{ccc} {\tilde{h}}_{t} & =\tanh(W_{h}\cdot \left[r_{t}⊙h_{t-1},x_{t}\right]+b_{h}),\; & //\;\mathrm{Candidate}\;\mathrm{hidden}\;\mathrm{state} \end{array}$$(10)$$\begin{array}{ccc} h_{t} & =\left(1-z_{t}\right)⊙h_{t-1}+z_{t}⊙{\tilde{h}}_{t},\; & //\;\mathrm{Final}\;\mathrm{hidden}\;\mathrm{state} \end{array}$$
where σ is the sigmoid function, tanh is the hyperbolic tangent activation, and ⊙ denotes element-wise multiplication. The reset gate $r_{t}$ determines how much of the previous hidden state to forget, while the update gate $z_{t}$ controls how much of the candidate state ${\tilde{h}}_{t}$ should be included in the new hidden state $h_{t}$.

### 2.2. Unsupervised Learning

Unsupervised methods do not rely on any human annotations. Instead, they extract segmentation-relevant information from intrinsic data patterns.

#### 2.2.1. Generating Adversarial Networks

Goodfellow et al. proposed GANs in 2014, which have become a major breakthrough in generative models [30]. The core idea of GANs is to train two networks through an adversarial process: a generator and a discriminator. These two networks compete with each other during the training process to improve the quality of the generated data. GANs, which are typically employed in unsupervised learning scenarios, integrate both generator and discriminator models. By utilizing two neural networks with opposing objectives, GANs achieve joint optimization through adversarial training [31]. The generator network is considered as a set of pretenders and the discriminator is responsible for classifying samples as true or false. By leveraging adversarial attacks on the discriminators, the generators deceive the discriminators, leading them to compete inaccurately. During the training process, the generators and the discriminators fight against each other: the generators try to produce data that look real, while the discriminators try to distinguish between real data and generated data. This process can be viewed as a minimax game, where the ultimate objective is for the generator to be capable of producing high-quality synthetic data that are sufficient to deceive the discriminator. Figure 5 shows the basic framework of GAN.

#### 2.2.2. Diffusion Model

A class of methods based on generative models, known as diffusion models (DMs), primarily generate data, including images, audio, and text. These models learn the distribution of data by modeling the inverse of the data generation process, i.e., the gradual generation of data from random noise. Diffusion models have received a significant deal of attention in recent years due to their excellent generation quality and versatility. We can divide the diffusion model’s working principle into two stages: the forward process and the reverse process. During the forward process, the model gradually introduces Gaussian noise into the data, ultimately transforming it into unstructured noise. This process usually consists of a series of predefined steps, each of which adds a certain amount of noise. In the reverse process, the model learns how to gradually recover the original data from the noise. This process essentially teaches the model the opposite of the forward process. Through training, the model learns to gradually remove the noise, eventually generating samples that match the distribution of the training data. Diffusion models have demonstrated extremely powerful generative capabilities in many domains, such as image generation, audio generation with super resolution, and so on. People widely recognize diffusion models for their high-quality and diverse generative capabilities. Compared to other generative models (e.g., GANs and VAEs), diffusion models are more stable during training and less prone to mode collapse. However, their main challenge is the high computational cost of the generative process, as multiple iterative steps are required to generate data from noise.

### 2.3. Reinforcement Learning

Reinforcement learning (RL) formulates segmentation as a sequential decision-making process. An agent learns to perform segmentation actions to maximize cumulative rewards, which are defined based on segmentation quality.

### 2.4. Deep Reinforcement Learning

Reinforcement learning is a theoretical framework that integrates psychological and engineering principles to investigate how intelligent agents interact with their environment to develop strategies that maximize expected rewards. Traditional reinforcement learning algorithms are predominantly designed for tabular scenarios, offering principled solutions for straightforward tasks. However, these algorithms encounter significant challenges when applied to highly complex domains, particularly those involving three-dimensional environments. Deep reinforcement learning emerges as a paradigm-shifting methodology that synergistically integrates reinforcement learning’s decision-making framework with deep neural networks’ representation capacity. This hybrid architecture capitalizes on deep learning’s hierarchical feature extraction to solve sequential decision problems in high-dimensional state spaces [32], effectively bridging the gap between perceptual understanding and strategic action planning in complex environments. Deep reinforcement learning provides a comprehensive framework for studying the interaction between learning, representation, and decision-making, offering a new set of research tools and a wide range of new hypotheses for neuroscience [33]. Most machine learning paradigms involve pre-collecting or constructing dataset labels and performing machine learning based on existing static data. In contrast, reinforcement learning is a typical representative of the closed-loop learning paradigm, using dynamic data and labels to bring feedback signals into the learning process (Figure 6). The emergent capability of deep reinforcement learning in constructing hierarchical state abstractions has positioned it as a principled methodology for sequential decision-making, where deep neural networks dynamically encode environmental observables while reinforcement learning orchestrates temporal credit assignment. The integration of these two advanced technologies is regarded as one of the most promising approaches to achieving artificial general intelligence [34].

### 2.5. Weakly Supervised Learning

Weakly supervised learning in deep learning is a method that utilizes limited or incompletely labeled data to train deep neural networks. This approach is particularly valuable when data labeling is costly or difficult. Weakly supervised learning operates on imperfectly annotated datasets [35], particularly in scenarios where merely image-level labels are accessible [36]. The related literature categorizes weakly supervised learning into three distinct categories.

More specifically, in incomplete supervision, common strategies include semi-supervised learning and active learning, where models leverage both labeled and unlabeled data. For instance, image classification tasks may use pseudo-labeling or consistency regularization techniques to exploit the large amount of unlabeled data. In the case of inaccurate supervision, label noise becomes a major challenge. To address this, methods such as robust loss functions, label correction algorithms, or noise-tolerant architectures have been developed. For example, in medical image analysis, misannotations are common due to subjective interpretation, and robust learning techniques help improve model reliability. Finally, uncertain supervision often arises in scenarios where labels are ambiguous or derived from weak sources, such as web-scraped data or user-generated tags. To handle this, researchers have explored probabilistic models or multiple instance learning (MIL), which allow learning from label distributions or ambiguous annotations.

The first type is incomplete supervision, also known as incomplete annotation. In this type of weak supervision, only a small portion of the training set is annotated [37]. The second type, known as inaccurate supervision, refers to weakly supervised object recognition using convolutional neural networks, where the labels in the dataset are not completely accurate. The third type is uncertain supervision, where only coarse labels exist [38]. Figure 7 shows how weakly supervised learning works.

#### Segment Anything Model

The segment anything model (SAM) is an advanced image segmentation model developed by Meta [39]. This model aims to automatically segment any object in any image without the need for pre-labeled datasets. The design goal of SAM is to quickly and accurately complete object segmentation tasks in images with minimal or even no labeled data, relying on its powerful learning ability and generalization capability. It employs efficient deep learning algorithms and large-scale pre-trained models, capable of handling various complex scenes and diverse image contents. SAM has a wide range of applications, including but not limited to autonomous driving, medical image analysis, intelligent surveillance, and augmented reality. By offering open-source code along with comprehensive technical documentation, SAM significantly enhances the development of image segmentation technology and provides robust tools and substantial support for associated research and applications.

It employs efficient deep learning algorithms and large-scale pre-trained models, capable of handling various complex scenes and diverse image contents.

Specifically, SAM is composed of three main components: a prompt encoder, an image encoder, and a mask decoder. The image encoder is based on a vision transformer (ViT), which processes the input image to produce a dense embedding $E_{I}$. The prompt encoder transforms various user prompts (e.g., points, boxes, or masks) into embeddings $E_{P}$. These two embeddings are fused within the mask decoder, which is a lightweight transformer decoder that predicts object masks in a single forward pass.

Let *I* be the input image and *P* be the prompt. The SAM pipeline can be summarized as follows:(11)$$\begin{array}{cc} E_{I} & =\mathrm{ViT}(I), \end{array}$$(12)$$\begin{array}{cc} E_{P} & =\mathrm{PromptEncoder}(P), \end{array}$$(13)$$\begin{array}{cc} M & =\mathrm{MaskDecoder}(E_{I},E_{P}), \end{array}$$
where *M* denotes the output segmentation mask. The model is trained with a combination of binary cross-entropy loss and Dice loss to improve both pixel-wise accuracy and mask quality.

A unique feature of SAM is its ability to handle multiple prompts in a unified framework, allowing segmentation to be guided by different types of user input. This promptable segmentation mechanism enables interactive segmentation and supports zero-shot generalization across various domains. Figure 8 shows the whole process of ultrasound image segmentation by SAM.

## 3. Deep Learning in Ultrasound Image and Video Segmentation

This paper categorizes deep learning-based methods for ultrasound image and video segmentation according to different learning paradigms, including (1) supervised learning-based methods, (2) unsupervised learning-based methods, (3) weakly supervised learning-based methods, and (4) deep reinforcement learning-based methods. A literature search was systematically conducted across multiple academic databases, including Google Scholar, PubMed, Web of Science, and Semantic Scholar. The search strategy involved a combination of keywords (e.g., “machine learning”, “deep learning”, “ultrasound image segmentation”, “ultrasound video segmentation”, “CNN”, “recurrent neural networks”, “deep reinforcement learning”), which were iteratively refined to capture a comprehensive set of relevant articles.

The search was limited to publications in English spanning the last 5–10 years to ensure that the review reflected the most recent advancements in the field. Papers were included based on the following labels: (1) studies that focus on ultrasound images or videos, including both conventional and deep learning-based segmentation methods; (2) articles that present segmentation techniques, evaluation methods (e.g., commonly used metrics, high-quality datasets), or comprehensive experimental evaluations specific to ultrasound imaging; (3) papers published in well-defined, peer-reviewed journals or conference proceedings with clear methodological details. Articles that did not meet the following criteria were excluded: (1) articles not directly related to ultrasound imaging segmentation or that focus solely on other imaging modalities; (2) non-English publications, reviews, commentaries, editorials, and conference abstracts without full-text availability; (3) duplicate records and studies that do not provide sufficient methodological details or experimental validation. Figure 9 illustrates the overall process of literature retrieval using a Preferred Reporting Items for Systematic Reviews and Meta-Analyses (PRISMA) flow diagram. Initially, about 25,000 documents were searched, and the search results were screened through several rounds of (1) removing duplicate papers to include the above criteria, (2) reading through the entire text to determine its relevance to the topic of this article, and excluding studies that were not pertinent to the research focus. Finally, about 97 articles were selected for use in this article, which systematically introduces the research on deep learning-based methods for ultrasound image and video segmentation and evaluates the model performance of individual studies, assessing the methodology and part of the high-quality dataset and generally ensuring the comprehensiveness of the deep learning-based methods for medical ultrasound images.

### 3.1. U-Net in Ultrasound Image and Video Segmentation

#### 3.1.1. Overview of Works

Recently, Vahid et al. [40] addressed this problem with a new U-Net-based architecture called fast and accurate U-Net for the medical image segmentation task. The proposed fast and accurate U-Net model primarily consists of four tuned 2D convolutional layers, a 2D transposed convolutional layer, and a batch normalization layer. There are four blocks in the codec path. The architectural performance was rigorously validated through dual evaluation protocols: (1) a dedicated segmentation task dataset measuring fetal head and abdominal circumference and (2) the publicly benchmarked HC18-Grand Challenge dataset for fetal head circumference quantification. This rapid network significantly improved processing time compared to U-Net, dilated U-Net, R2U-Net, attention U-Net, and MFP U-Net. Amiri et al. [41] studied the impact of fine-tuning different layers of a pre-trained U-Net on breast ultrasound image segmentation. Compared to fixing the contraction part and adjusting the expansion part, adjusting the contraction part while fixing the expansion part produced significantly better results. In the same year, the authors proposed using one U-Net to detect lesions before performing breast ultrasound image segmentation and then using another U-Net to segment the detected areas. This approach improved the average Dice score by 1.8%. For images with an original Dice score below 70%, the average Dice score increased by 14.5% [42]. Cheng et al. [43] explored the best transfer learning techniques for accurate and fast image segmentation using U-Net. The study employed two transfer learning methods: first, they built U-Net using a pre-trained VGG16 model (V-Unet); second, they trained U-Net using a grayscale natural salient object dataset. The Dice coefficient of 0.8 achieved a satisfactory level. At the same time, Chen et al. [44] suggested a simple but powerful Nested U-net (NU-net) for accurately separating breast tumors. NU-net uses U-Nets with shared weights to create a strong representation of breast tumors. NU-net reduces the network’s sensitivity to input images of different scales and further enhances the representation capability of objects or regions. Xu et al. [45] suggested a multi-scale feature extraction and fusion network (MEF-UNet) for ultrasound image segmentation. It has three parts: a selective feature extraction module, a contextual information storage module, and a multi-scale feature fusion module. Neural Architecture Search (NAS) has made significant progress in improving image classification accuracy. Recently, some work has attempted to extend NAS to image segmentation. Weng et al. [46], inspired by the U-net architecture and its successful variants applied to various medical image segmentation tasks, proposed NAS-Unet. The proposed architecture employs symmetrical DownSC (Downsampling Separable Convolution) and UpSC (Upsampling Separable Convolution) blocks within a U-shaped topology, demonstrating superior segmentation accuracy with significantly reduced parameter complexity compared to standard U-Net architectures and their derivatives. Ultrasound image annotation heavily relies on skilled physicians, such as in transvaginal ultrasound (TVUS), commonly used for diagnosing infertility, where doctors manually annotate the size and shape of ovaries and follicles. Li et al. [47] proposed a new composite network, CR-Unet, to simultaneously segment ovaries and follicles in TVUS. The proposed architecture incorporates a recurrent neural network within a U-Net framework to capture multi-scale spatial dependencies and long-range contextual patterns. Experimental evaluations on a cohort of 219 patients (3204 TVUS images) demonstrated segmentation accuracy with mean DSC scores of 0.912 for ovarian structures and 0.858 for follicular regions. To aid diagnosis, the automatic segmentation of regions of interest (ROIs) in ultrasound images has become an essential component of computer-aided diagnosis (CAD). Wang et al. [48] proposed a Scale Attention Convolution module, which constructs MSAC-Unet based on U-net, using MSAC instead of standard convolutions in each encoder and decoder for segmentation. MSAC-Unet did the best job of segmenting three datasets: two thyroid nodule datasets (TND-PUH3 and DDTI) and one brachial plexus dataset (NSD), and all three had Dice coefficients of 0.822, 0.792, and 0.746. MSAC-Unet significantly improved segmentation accuracy, with more reliable ROI edges and boundaries, and reduced the number of incorrectly segmented ROIs in ultrasound images. Inan et al. [49] proposed a new AI decision support system for the automatic segmentation and classification of thyroid nodule types. This system is based on a hybrid deep learning approach, using ResUNet++ for automatic thyroid nodule segmentation and combining DenseNet and ResNet for classification. Table 1 summarizes the data sets, models used, areas of focus, and model performance of the literature reviewed above.

#### 3.1.2. Assessments

U-Net has demonstrated significant performance improvements in medical ultrasound image and video segmentation. Fast and accurate architectures, transfer learning, nested U-Net, multi-scale feature extraction and fusion networks, and composite networks are among the advancements that have significantly enhanced the accuracy and efficiency of segmentation using U-Net. These improvements not only reduce processing time but also enhance robustness to input images of different scales, improving Dice coefficients, especially in specific tasks like breast ultrasound, fetal head circumference measurement, and thyroid nodule segmentation. The results show that U-Net and its variations are very good at dealing with different kinds of complex lesions. This makes them a strong technical support for computer-aided diagnosis and automated segmentation in ultrasound imaging.

### 3.2. Fully Convolutional Neural Network in Ultrasound Image and Video Segmentation

#### 3.2.1. Overview of Works

Fully convolutional network-based methodologies continue to advance precision medicine in medical ultrasound analysis. For lymph node characterization, Zhang et al. [50] pioneered a cascaded Coarse-to-Fine Stacked FCN (CFS-FCN), utilizing multi-stage FCN modules to automate lymph node delineation, while Villa et al. [51] developed an FCN-driven bone interface localization algorithm, outperforming manual segmentation and Phase Symmetry Confidence (CPS) benchmarks. In breast cancer diagnostics, Xing et al. [52] innovatively integrated FCNs with GANs to propose the Semi-Pixel Cycle GAN (SPCGAN), elevating lesion segmentation performance from 0.90 to 0.93 DSC. Hu et al. [53] addressed speckle noise challenges through a dilated FCN (DFCN) coupled with Phase-Based Active Contour models, leveraging deep dilated convolutions (rates 6–24) and batch normalization to enhance tumor-background differentiation in breast ultrasound.

Expanding the FCN’s multitask capabilities, Li et al. [54] incorporated an attention-guided feature selection mechanism and an elliptical ROI pooling layer, enabling simultaneous fetal biometric measurement (OFD/BPD) and anatomical segmentation. Qian et al. [55] achieved state-of-the-art breast ultrasound segmentation via quad-branch dilated convolutions (rates 1/3/5/7) for multiscale context aggregation, refined by Laplacian edge correction. For prostate cancer analysis, Feng et al. [56] constructed a multi-stage FCN architecture encoding spatial–boundary correlations, attaining 94.90% DSC in gland segmentation. In dynamic imaging, Xu et al. [57] proposed the Local–Global Reciprocal Network (LGRNet), synergizing Cyclic Neighborhood Propagation (CNP) for spatiotemporal context modeling with the Hilbert Selective Scan (HilbertSS) for noise suppression, achieving 89.80% IoU and 96.49% recall in uterine fibroid video analysis through hybrid DenseNet-ResNet classification modules. Table 2 summarizes the data sets, models used, areas of focus, and model performance of the literature reviewed above.

#### 3.2.2. Assessments

FCNs have significant advantages and some drawbacks in medical ultrasound image segmentation. The primary advantages include their ability to perform pixel-level predictions throughout the entire process, thereby simplifying segmentation and enhancing the efficiency of both training and inference. Additionally, they can effectively extract spatial features from images via convolution operations, making them well suited for handling large volumes of image data. Moreover, their high degree of flexibility allows them to adapt to a variety of segmentation tasks. However, FCNs also face some disadvantages, such as their limited capacity to capture fine-grained features, particularly in complex and boundary-blurred ultrasound images. Additionally, FCNs are weak in modeling long-range dependencies, which limits their performance in tasks requiring global contextual information. Future directions for FCNs in medical ultrasound image segmentation could encompass incorporating attention mechanisms to enhance detail capture, integrating multi-scale features to boost accuracy, and combining with other networks such as RNNs or transformers to address limitations in long-range dependency modeling. These advancements would collectively improve overall segmentation performance and application effectiveness.

### 3.3. Recurrent Neural Networks in Ultrasound Image and Video Segmentation

#### 3.3.1. Overview of Works

Due to the need to not only remember relevant information but also actively forget irrelevant information, LSTM is one of the most widely used architectures in recurrent neural networks. In recent years, this architecture has been increasingly utilized to address challenges in automatic ultrasound image and video segmentation, such as the lack of distinct boundaries and significant shape variations in ultrasound images. Pan et al. [58] proposed an automatic breast tumor segmentation method based on bidirectional LSTM and spatial-channel attention modules integrated into a fully convolutional network. The Dice Similarity Coefficient (DSC), Recall, Precision, and Hausdorff Distance (HD) of this method were 0.8178, 0.8067, 0.8292, and 11.1367, respectively. Transperineal ultrasound (TPUS) serves as the primary imaging modality for evaluating pelvic floor disorders. Noort et al. [59] proposed a U-net-like neural network with several convolutional LSTM layers to automatically segment the levator ani muscle (LAM) in TPUS volumes, achieving human-level performance in the segmentation task. To address the issue of incomplete boundaries in prostate ultrasound images, Yang et al. [60] proposed using a recurrent neural network to learn shape priors, employing a multi-view fusion strategy to integrate shape predictions from different views. This method outperformed the state-of-the-art CNNs and FCNs. Meanwhile, Webb et al. [61] proposed a four-stage trained convolutional LSTM model based on DeepLabv3C for segmenting thyroid ultrasound images, achieving average intersection scores of 0.427, 0.533, and 0.739 for cysts, nodules, and the thyroid. To avoid the time-consuming and labor-intensive manual intervention in ultrasound image segmentation, Horng et al. [62] proposed a convolutional neural network framework called DeepNerve, which is based on U-Net and combines Masktrack and LSTM for locating and segmenting the median nerve. The experimental results showed that the generated average Dice measure, precision, recall, and F-score were 0.8975, 0.8912, 0.9119, and 0.9015, respectively. Birth defects and associated mortality are among the foremost causes of infant death. Devisri et al. [63] proposed a method for fetal growth analysis using head circumference biometrics from ultrasound images through optimal segmentation and hybrid classifiers. This method uses U-Net to gather important features, enhances the boosting algorithm to choose the best features, and combines bidirectional LSTM and a convolutional neural network (B-LSTM-CNN) for analyzing fetal growth. Compared to LSTM, gated recurrent units (GRUs) require less memory and exhibit similar performance. Anas et al. [64] developed a hybrid CNN-GRU architecture for prostate ultrasound segmentation, where convolutional layers extracted spatial features while gated recurrent units captured temporal dependencies. Their experimental protocol utilized 2238 labeled transrectal ultrasound images for training, with 637 and 1017 images allocated for validation and testing, respectively, achieving a Dice coefficient of 0.93. Notably, early breast cancer detection remains clinically critical due to its profound impact on therapeutic efficacy and patient survival rates. Khdhir et al. [65] came up with a new hybrid method for breast cancer segmentation that uses the Gray-Level Co-occurrence Matrix (GLCM) for feature extraction and gated recurrent units for classification after preprocessing. Table 3 summarizes the data sets, models used, areas of focus, and model performance of the literature reviewed above.

#### 3.3.2. Assessments

Because they can pick up on temporal dependencies and contextual information across image sequences, RNNs have shown a lot of promise in the field of medical ultrasound image and video segmentation. These networks excel in scenarios where the spatial–temporal coherence of the data is crucial for accurate segmentation. One of the primary advantages of RNNs is their capacity to integrate temporal information, which is particularly beneficial in medical ultrasound imaging. Unlike static images, ultrasound data often comprise sequential frames that capture dynamic biological processes. By leveraging RNNs, segmentation algorithms can utilize the continuity and evolution of these frames, leading to more accurate and consistent segmentation results. However, the application of RNNs in medical ultrasound segmentation is not without challenges. The sparsity of manual annotations available for training poses a critical issue. Medical annotations are often labor-intensive to produce, and obtaining a comprehensive dataset with dense annotations can be impractical. To address this, advanced strategies such as label propagation and sophisticated loss functions have been employed to enhance the training process despite sparse annotations.

### 3.4. Generating Adversarial Networks and Diffusion Models in Ultrasound Image and Video Segmentation

#### 3.4.1. Overview of Works

Liang et al. [67] looked into an autoencoder generative adversarial network that combined the best features of GANs and variational autoencoders (VAEs) to make realistic images for medical thyroid ultrasound image enhancement. They tackled the issue of mode collapse in GANs, which results in less varied images. Conversely, mode collapse does not affect VAEs, but they do tend to produce blurry images. The test results showed that the images created mimicked real ultrasound features and thyroid tissue, improving training and helping U-Net models perform better in segmenting images. Liang et al. [68] proposed an image synthesis framework based on GANs. Recent advances in synthetic medical imaging have introduced novel approaches for enhancing anatomical realism. Bargsten et al. [69] developed SpeckleGAN, a specialized GAN framework that synthesizes domain-specific speckle patterns through learnable noise layers, particularly valuable for ultrasound modality simulation. This architecture uniquely integrates trainable speckle generators as differentiable network components, enabling physics-informed texture synthesis. Meanwhile, Fatima et al. [70] proposed a new method for the automatic segmentation of two-dimensional echocardiographic images. This method uses a generative adversarial network (Pix2Pix GAN), which consists of the PatchGAN model as the discriminator and the U-Net model as the generator, to accurately segment cardiac structures. Simultaneously, based on the diffusion model, Tang et al. [71] proposed a multi-layer global-context cross-consistency (MGCC) framework, which uses images generated by a latent diffusion model (LDM) as unlabeled images for semi-supervised learning. This framework uses an LDM to generate synthetic medical images, reducing the workload of data annotation and addressing privacy issues associated with medical data collection. Stojanovski et al. [72] proposed a new denoising diffusion probabilistic model guided by cardiac semantic labels to address the challenge of limited labeled data in the medical imaging field, particularly in ultrasound image analysis and cardiac segmentation tasks. They demonstrated an innovative medical image synthesis and analysis process, showing that synthetic images can effectively replace real data in training deep learning models. Katakis et al. [73] proposed a method for generating musculoskeletal ultrasound images using a diffusion model, demonstrating that synthetic data play a crucial role in the final performance of models and can improve deep learning systems in musculoskeletal ultrasound. Stevens et al. [74] proposed a joint posterior sampling framework that combines two independent diffusion models to simulate the distribution of clean and hazy ultrasound images in a supervised manner. The proposed dehazing method works well to get rid of haze while keeping signals from tissues that are weakly reflective, as shown by experiments conducted with living things and in the lab. Yao et al. [75] proposed a semi-supervised ultrasound image segmentation method called the Dual-Frequency Cascade Graph (DFCG) model. This method uses a latent diffusion model to make fake medical images to make data annotation easier. It also combines Fourier frequency domain space with a multi-scale attention mechanism to lower the effect of scattering noise. Table 4 summarizes the data sets, models used, areas of focus, and model performance of the literature reviewed above.

#### 3.4.2. Assessments

These methods contribute to improving the performance of segmentation models by generating high-resolution synthetic images with realistic features. GANs have demonstrated their powerful capabilities in image synthesis, producing realistic ultrasound images and enhancing structural details, while diffusion models have improved the performance of segmentation models on real images by generating large amounts of synthetic data. However, challenges remain, including GANs’ susceptibility to mode collapse during training and the limited diversity of generated images. Diffusion models, on the other hand, require substantial computational resources, and the quality of the generated images is highly dependent on the initial data quality. Future research trends may focus on improving the quality and diversity of generated images, reducing mode collapse during training, and exploring the generalizability of these models across different types of medical images and applications. Additionally, we expect to further enhance the accuracy and efficiency of medical image segmentation by integrating GANs and diffusion models with other advanced deep learning techniques.

### 3.5. Deep Reinforcement Learning in Ultrasound Image and Video Segmentation

#### 3.5.1. Overview of Works

For the segmentation of the prostate in transrectal ultrasound images, Sahba et al. [76] introduced a novel method that aims to find suitable local values for sub-images and extract the prostate. It comprises an offline phase wherein a reinforcement learning agent is trained on a dataset consisting of multiple images along with their manually segmented counterparts. Wang et al. [77] proposed a medical image segmentation framework based on online reinforcement learning. The model adapts not only to the defined objective functions but also to user intentions and prior knowledge. Detailed validation results in the paper indicate that this model can significantly reduce user interaction while maintaining segmentation accuracy and consistency. Mathews et al. [78] created a new unsupervised reinforcement learning (RL) framework with new rewards to help unsupervised learning. This way, the manual labeling of ultrasound videos does not have to be as time-consuming and impractical, and the framework can be used more easily as a triage tool in emergency departments and for telemedicine. Ning et al. [79] presented a reinforcement learning (RL)-based autonomous robotic ultrasound imaging system. The proposed system and framework were designed to control the US probe, enabling it to perform fully autonomous imaging of soft, mobile, and unlabeled targets, using only a single RGB image of the scene.

#### 3.5.2. Assessments

DRL for medical ultrasound image and video segmentation optimizes segmentation strategies through trial-and-error methods and reward mechanisms. DRL models adapt well to complex imaging scenarios and user-defined goals, improving segmentation accuracy and reducing user interaction. These methods excel at handling local value extraction, context adaptation, and reducing manual annotation efforts. However, challenges remain, including high computational costs, potential instability during training, and the need for large and diverse training datasets. In the future, people will probably focus on making DRL models more reliable and usable across a wider range of ultrasound types, combining them with other cutting-edge methods to improve segmentation performance, and reducing the need for human intervention even more. These advancements will help address current limitations and broaden the applicability of DRL in clinical settings.

### 3.6. Weakly Supervised Learning in Ultrasound Image and Video Segmentation

#### 3.6.1. Overview of Works

Girum et al. [80] came up with a fast, interactive, weakly supervised deep learning method for medical ultrasound image segmentation to get around the problems with annotated datasets and the time-consuming nature of manual annotation. GANs were used to predict pseudo-contour landmark priors. Then, CNNs were used to improve the predicted priors (i.e., contour suggestions). The experimental results showed that this method achieved a Dice coefficient of 0.97 in colorectal ultrasound image segmentation. Additionally, Li et al. [81] proposed a two-step deep learning framework for breast ultrasound image segmentation. The first step is a semi-supervised semantic segmentation framework that decomposes four anatomical structures: the fat layer, breast layer, muscle layer, and chest cavity layer. The second step is to segment breast tumors at the image-level labels in a weakly supervised learning environment. The Dice coefficients for the four anatomical structures were 83.0 ± 11.8%, 84.3 ± 10.0%, 80.7 ± 15.4%, and 91.0 ± 11.4%, respectively. Regarding breast tumor segmentation, Li et al. [82] proposed a network based on a CNN backbone and spatial pyramid module, achieving a maximum Dice coefficient of 73.5 ± 18.0% in a weakly supervised learning scenario. Table 5 summarizes the data sets, models used, areas of focus, and model performance of the literature reviewed above.

#### 3.6.2. Assessments

Weakly supervised learning has shown a lot of promise in medical ultrasound image and video segmentation by lowering the need for large datasets with lots of annotations while still maintaining strong segmentation performance. Achievements in this area include the development of novel frameworks that facilitate unsupervised learning and reduce the need for tedious manual labeling, thereby enhancing the utility of ultrasound imaging tools in emergency departments and telemedicine. These methods use advanced structures like attentional encoders and bi-directional long short-term memory networks to project high-dimensional images into lower-dimensional latent spaces more efficiently. This makes it easier to classify and divide images. However, challenges persist, including limited generalization to various imaging conditions and the potential need for high-quality initial data to train effective models. Future trends will likely focus on improving weakly supervised techniques to better manage different and changing imaging situations. This will include combining these methods with reinforcement learning to make imaging systems more independent and reduce the need for manual help. This technical convergence between attention-aware feature distillation and uncertainty-quantified pseudo-label generation paves the way for expanding weakly supervised applications into real-time clinical decision support systems, overcoming traditional barriers in annotation fidelity.

### 3.7. Segment Anything Model in Ultrasound Image and Video Segmentation

#### 3.7.1. Overview of Works

The development of SAM has expanded its applications beyond natural images. Tu et al. [84] proposed the breast ultrasound segment anything model (BUSSAM), which migrates the SAM model to the field of breast ultrasound image segmentation through adapter technology. They achieved more accurate segmentation results on ultrasound image datasets by designing a lightweight CNN image encoder and introducing position and feature adapters to further fine-tune the branch. In the segmentation of tumors and cancerous regions in breast ultrasound images, Guo et al. [85] proposed the ClickSAM model, which fine-tunes the segment anything model using click prompts to improve the accuracy of ultrasound image segmentation. Pandey and others discovered that using YOLO + SAM for breaking down aortic images worked better than regular U-Net models, especially with POCUS images that had a lot of noise and small areas of interest. At the same time, research combining SAM with other models shows that the model is comparable to other deep learning models. Pandey et al. [86] compared the YOLOv8, YOLOv8 + SAM, and YOLOv8 + HQ-SAM models, demonstrating significant advantages in multimodal medical image segmentation. The Dice scores on the short-axis aorta ultrasound dataset were 0.5064, 0.769, and 0.7722, respectively. Zhao et al. [87] suggested a SAM-based weakly supervised framework that does a better job of segmenting nodules in clinical ultrasound images by creating fake labels and using a cross-teaching strategy that takes uncertainty into account. The application of SAM to ultrasound video The segmentation is equally surprising. Deng et al. [88] introduced MemSAM, adapting SAM to echocardiography videos via a spatiotemporal memory that encodes temporal cues and refines features with predicted masks to address speckle noise and ambiguous boundaries. It achieves state-of-the-art performance on two datasets with limited annotations, rivaling fully supervised methods. This work pioneers SAM’s medical video adaptation through noise-resilient, temporally consistent prompting. Kim et al. [89] developed MediViSTA, a parameter-efficient SAM adaptation for echocardiography videos, combining spatial frequency fusion and temporal adapters to enhance 3D medical data processing. It outperforms SOTA methods across three datasets, achieving superior generalization (2.15% Dice) without prompts, advancing cardiac segmentation accuracy and temporal consistency. Pang et al. [90] proposed BaS, a real-time breast lesion segmentation model for ultrasound videos, employing Stem modules and BaSBlock for inter-/intra-frame analysis. With variants BaS-S (accuracy) and BaS-L (speed), it outperforms SOTA in efficiency and precision on resource-limited devices, enabling practical clinical deployment.

#### 3.7.2. Assessments

The adaptation of the SAM to ultrasound image and video segmentation has demonstrated notable progress but also highlighted critical challenges. Recent studies reveal SAM’s adaptability through lightweight architectural modifications, such as BUSSAM’s CNN encoder and position adapters and MediViSTA’s spatiotemporal adapters, enabling the efficient processing of noisy, low-contrast ultrasound data while maintaining parameter efficiency. Innovations like ClickSAM’s click-based prompts and MemSAM’s temporal-aware memory mechanisms further enhance segmentation accuracy by addressing speckle noise and ambiguous boundaries, achieving performance comparable to fully supervised methods with minimal annotations. Weakly supervised frameworks, such as Zhao et al.’s pseudo-labeling with uncertainty-aware cross-teaching, underscore SAM’s potential to reduce annotation dependency. However, challenges persist: SAM’s inherent 2D architecture necessitates complex 3D adaptations (e.g., MediViSTA’s temporal modules) for video analysis, increasing computational overhead. Multi-stage pipelines like YOLO + SAM, though effective for small regions in noisy POCUS images, suffer from high inference costs, limiting real-time deployment. Additionally, prompt-dependent methods (e.g., ClickSAM) hinder fully automated workflows. To bridge these gaps, recent efforts prioritize efficiency and scalability: BaS introduces inter-/intra-frame modules for real-time lesion segmentation on edge devices, while MemSAM leverages temporal consistency to minimize error propagation. Future directions should focus on 3D-native SAM extensions for volumetric ultrasound, self-supervised training to eliminate prompts, and lightweight architectures balancing accuracy and speed. Collectively, these advancements position SAM as a versatile foundation for medical imaging, yet its clinical translation demands further optimization in computational efficiency and dimensional adaptability.

### 3.8. Datasets

The implementation of deep learning relies on large-scale data, which necessitates a high quantity and quality of medical images in the field of medical imaging deep learning research. To ensure that data are suitable for target tasks, it is often required to collect medical images from multiple medical centers and different types of imaging equipment, thereby guaranteeing that the trained deep learning models possess excellent generalization performance. Additionally, obtaining large-scale medical images in clinical settings is challenging and usually requires manual annotation by skilled clinicians, which demands considerable time and effort. For image segmentation, high-quality datasets are crucial for achieving more precise segmentation. Therefore, high-quality public datasets are of paramount importance. This section will introduce some high-quality datasets of ultrasound images and videos. We integrated the statistical dataset into Table 6, which describes the corresponding parts of the dataset, the quantities, and the URLs.

## 4. Evaluation Metrics

This chapter will mainly introduce a summary of experimental results indexes based on ultrasonic image and video segmentation methods so that readers can understand the meaning of each index. The performance of various indicators clearly measure the experimental results of each research method. Therefore, it is very important to introduce the common evaluation indicators.

### 4.1. Medical Image Segmentation Evaluation Methods

Good segmentation results require effective evaluation methods to assess the segmentation results, enabling the differentiation of segmentation algorithms or the adjustment of algorithm parameters based on the evaluation results. More importantly, only after evaluating the algorithms to select the optimal segmented images can the next step of processing be carried out. With the development of deep learning models, clinicians, particularly those in radiology and pathology, strive to integrate deep learning-based medical image segmentation methods into their clinical routines as Clinical Decision Support (CDS) systems to aid in diagnosis, treatment, risk assessment, and the reduction of time-consuming examination processes. Given the direct impact on diagnostic and treatment decisions, the correct and robust evaluation of algorithms is crucial. Therefore, this section reviews common performance evaluation methods for ultrasound (BUSI) image segmentation, with the metrics used referenced in Table 7. Table 8 shows the indicator performance of different technical paths in the same Breast Ultrasound Images Dataset. We explain the indicators so that readers can understand the meaning of each indicator more clearly.

#### 4.1.1. Dice Similarity Coefficient

A segmentation algorithm uses DSC, a symmetric similarity metric, to measure the overlap between its results and the ground truth. Most publications regard DSC as the primary metric for validation and performance interpretation in medical image segmentation.(14)$$DSC=\frac{2\cdot TP}{2\cdot TP+FP+FN}$$

#### 4.1.2. Jaccard Similarity Index

Medical ultrasound imaging uses the Jaccard similarity index (JSI) to assess the overlap between automated and manual segmentation results. It measures the accuracy and consistency of the segmentation algorithm by calculating the ratio of the intersection to the union of the two segmentation results. A high JSI value indicates a high degree of agreement between the automated and manual segmentation results, and vice versa.(15)$$JSI=\frac{TP}{TP+FP+FN}$$

#### 4.1.3. Hausdorff Distance

Medical ultrasound imaging uses the Hausdorff Distance to evaluate the shape similarity between two segmentation results. It quantifies the boundary differences of the segmentation results by measuring the distance between the farthest points of two point sets. This is crucial for evaluating the accuracy of automated segmentation algorithms, as it can detect extreme deviations at the boundaries.(16)$$H\left(A,B\right)=\max\left\{\underset{a\in A}{\sup}\underset{b\in B}{\inf}d\left(a,b\right),\underset{b\in B}{\sup}\underset{a\in A}{\inf}d\left(b,a\right)\right\}$$

#### 4.1.4. Precision

Since segmentation can be considered a classification problem for each pixel, Precision and Recall are also used as metrics. They can reflect the false positive and false negative pixels in the segmentation results.(17)$$Precision=\frac{TP}{TP+FP}$$

#### 4.1.5. Accuracy

Accuracy in ultrasound image segmentation refers to the percentage of pixels correctly classified as either the target region (e.g., lesion area) or the background.(18)$$Accuracy=\frac{TP+TN}{TP+TN+FP+FN}$$

#### 4.1.6. Recall

Recall in ultrasound image segmentation refers to the percentage of pixels accurately predicted as part of the target region. A high Recall indicates that the model has a strong ability to recognize the target region, with few false negatives (target region pixels incorrectly identified as background).(19)$$Recalll=\frac{TP}{TP+FN}$$

#### 4.1.7. F1-Score

In ultrasound image segmentation, the F1-score synthesizes the performance of both Precision and Recall, serving as a balanced metric between the two. It is particularly useful when there is a need to simultaneously consider the accuracy and completeness of the model. The F1-score provides a more comprehensive evaluation metric under these circumstances.(20)$$F1=2\cdot \frac{Precision\cdot Recall}{Precision+Recall}$$

## 5. Discussion and Conclusions

### 5.1. Discussion

In recent years, medical imaging, encompassing ultrasound images and videos, has seen the widespread application of deep learning techniques. We categorized the typical methods into four major classes, providing an introduction and a brief summary of related work. Based on a summary of existing research achievements, this paper outlined the issues, challenges, and potential research directions in the field. In the segmentation of medical ultrasound images and videos, convolutional neural network models such as U-Net, RNN, and FCN each have their own advantages and disadvantages. U-Net, with its symmetric encoder–decoder structure and skip connections, effectively captures multi-scale features and performs well in handling complex medical images. However, its high computational resource demands and dependence on large amounts of high-quality annotated data are significant drawbacks. While RNNs demonstrate significant advantages in processing ultrasound image sequences and capturing temporal information through mechanisms like long short-term memory, their application in static image segmentation remains limited. FCNs employ a unique architecture that enables them to predict image details at the pixel level throughout the entire process. They work well with large images, but they struggle a bit with picking up small features. Future trends in the development of these models include the integration of multi-modal data, the introduction of attention mechanisms, and improvements in model lightweight design to enhance segmentation performance and computational efficiency, better meeting clinical application needs. In clinical medical imaging, there are various imaging modalities, such as MRI, X-rays, PET, and CT. We employ different diagnostic imaging methods based on different cases. Ultrasound imaging is considered the preferred method due to its low cost and lack of radiation. MRI or CT, while providing rich texture information, are time-consuming and expensive. To address the processing challenges posed by multiple resources, cross-channel transfer learning is a promising research direction.

In the field of medical ultrasound image processing, the quality and quantity of datasets are crucial. High-quality annotated datasets can significantly improve the training efficiency and segmentation accuracy of deep learning models. However, annotating ultrasound images often requires specialized knowledge, making the manual annotation process time-consuming and prone to bias. The sensitivity and privacy concerns of medical data also pose challenges in obtaining large annotated datasets. Therefore, efficiently utilizing limited annotated data and enhancing the generalization ability of models are current research focuses. Generative models, especially diffusion models, have shown enormous potential in ultrasound image segmentation. Diffusion models, by gradually adding noise and learning the reverse process to generate high-quality image data, can generate realistic medical images under unsupervised or weakly supervised conditions, aiding model training. The introduction of generative models, such as diffusion models, not only enriches the methods for ultrasound image processing but also provides new solutions for addressing the lack of annotated data and improving model performance. In medical ultrasound images, various pathological changes exist, which can be minute and difficult to detect. SAM needs to have high sensitivity and specificity to detect and segment these pathological areas. By introducing reinforcement learning and adaptive training methods, SAM can continuously learn and improve its segmentation performance in dynamic and real-time environments. This can enhance the model’s applicability in various clinical scenarios. Future research can further explore the performance and optimization strategies of generative models in different ultrasound imaging application scenarios, thereby promoting the development of medical image processing technology.

### 5.2. Application

Recent advancements in deep learning have spurred the translation of research algorithms into commercially viable segmentation solutions across diverse clinical imaging modalities. For instance, GE Healthcare’s innovative project, SonoSAMTrack, leverages NVIDIA-powered deep learning to accurately delineate anatomical structures and lesions in ultrasound images, demonstrating robust performance on datasets ranging from adult cardiac to fetal head imaging, as well as musculoskeletal pathologies. In parallel, GE’s LOGIQ E20 ultrasound system integrates a dual-engine design that couples advanced signal processing with AI-based lesion recognition, streamlining segmentation workflows and reducing the need for extensive manual annotation. Philips and Siemens have similarly incorporated deep learning into their platforms: Philips’ ISAI platform and Siemens’ AI-Rad Companion suite both exemplify how automated segmentation is being embedded into routine clinical practice to improve diagnostic accuracy and workflow efficiency. Moreover, several emerging startups (e.g., Koios Medical and DiA Imaging) have introduced segmentation solutions that are optimized for minimal training data and high adaptability across different scanning protocols and devices, further reinforcing the trend toward clinically integrated, AI-driven tools. Collectively, these initiatives not only underscore the technological maturity of deep learning in segmentation but also illustrate a paradigm shift from isolated research prototypes to integrated, multi-vendor solutions that are now being validated in real-world clinical environments and have gained regulatory clearances, thereby paving the way for broader adoption in precision diagnostics and treatment planning. Moreover, they facilitate the detection of small lesions in breast or thyroid ultrasound, reducing operator dependence and improving early detection rates. Alongside these commercial developments, open-source toolkits and large-scale public datasets (e.g., CAMUS for cardiac ultrasound, BUSI for breast lesions) empower researchers and innovators to refine algorithms, while international AI competitions (e.g., at MICCAI) continue to push the envelope for real-time and 3D/4D segmentation capabilities. Efforts to address data variability, privacy concerns, and regulatory standards—via federated learning, rigorous clinical validation, and adherence to guidelines—are further accelerating the transition of deep learning–driven ultrasound segmentation from research labs to everyday clinical practice.

### 5.3. Challenges and Opportunities

Ultrasound image and video segmentation using deep learning faces several significant challenges. First, the inherent characteristics of ultrasound data—such as speckle noise, low contrast, and blurry boundaries—make accurate segmentation difficult. Second, the variability in image quality across different devices and operators introduces inconsistencies that hinder model generalization. Additionally, obtaining high-quality, pixel-level annotations for training is time-consuming and requires expert knowledge, which limits the availability of large annotated datasets. Furthermore, real-time segmentation in video sequences demands that models that are not only accurate but also computationally efficient. Addressing these challenges is crucial for the successful deployment of deep learning models in clinical ultrasound applications. This section presents solutions to these challenges under different learning paradigms.

### 5.4. Conclusions

In our review, we systematically collected and analyzed nearly 90 articles focusing on deep learning-based methods for ultrasound image and video segmentation. The results from these studies indicate that deep learning models—especially those based on UNet and its variants—consistently outperform traditional segmentation methods. For instance, many of the selected studies reported an average Dice similarity coefficient of approximately 0.85, which is an improvement of over 15% compared to conventional approaches. Furthermore, evaluation metrics such as Intersection over Union (IoU) and sensitivity were generally higher in deep learning models, demonstrating their robustness in handling the inherent noise and variability in ultrasound imaging.

Additionally, our analysis revealed that while deep learning techniques show promising performance, challenges remain in terms of model generalization, especially when applied to multi-center datasets with varying imaging protocols. Several studies noted that the performance drop on external validation datasets highlights the need for standardized data acquisition and preprocessing methods. Moreover, the integration of these algorithms into clinical workflows—although promising—requires further validation to ensure that automated segmentation outputs are reliable enough for clinical decision-making.

These findings not only substantiate our conclusions regarding the significant performance advantages of deep learning in ultrasound segmentation but also underline critical areas for future research. Standardizing evaluation metrics and expanding the availability of high-quality annotated datasets will be essential steps toward improving model generalizability and clinical applicability.

## Figures and Tables

**Figure 3 sensors-25-02361-f003:**
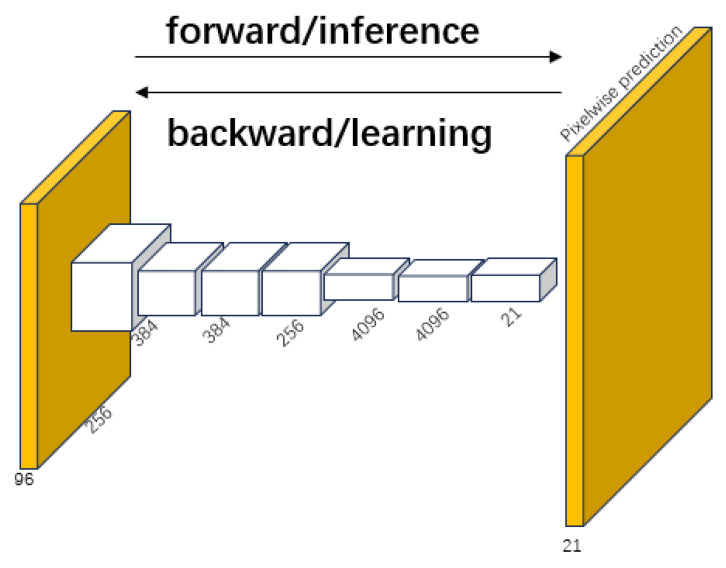
FCN’s resolution-preserving fully convolutional layers.

**Figure 4 sensors-25-02361-f004:**
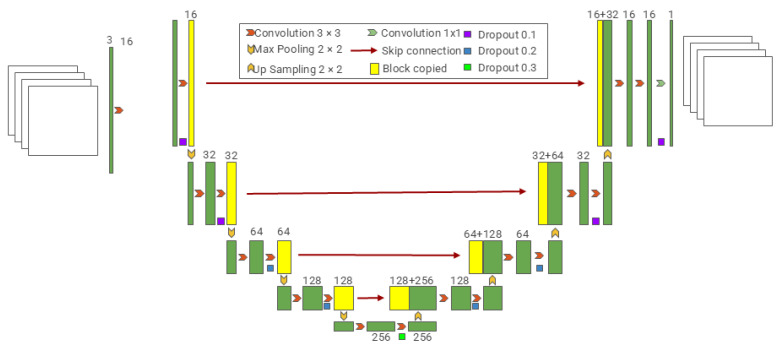
U-Net’s skip-connected encoder–decoder for medical segmentation.

**Figure 5 sensors-25-02361-f005:**
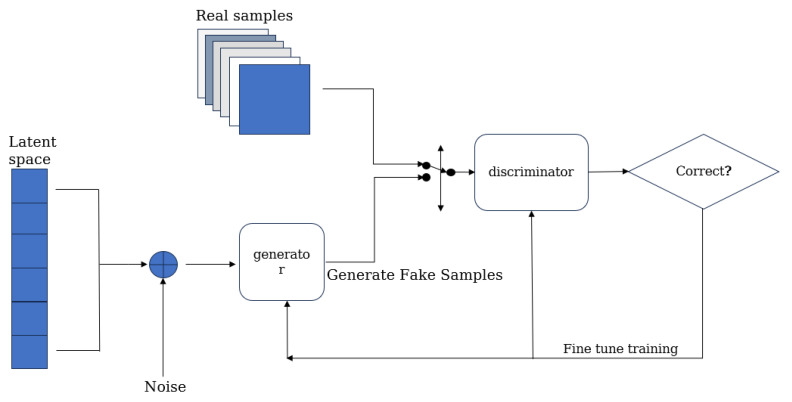
GAN’s adversarial generator–discriminator framework.

**Figure 6 sensors-25-02361-f006:**
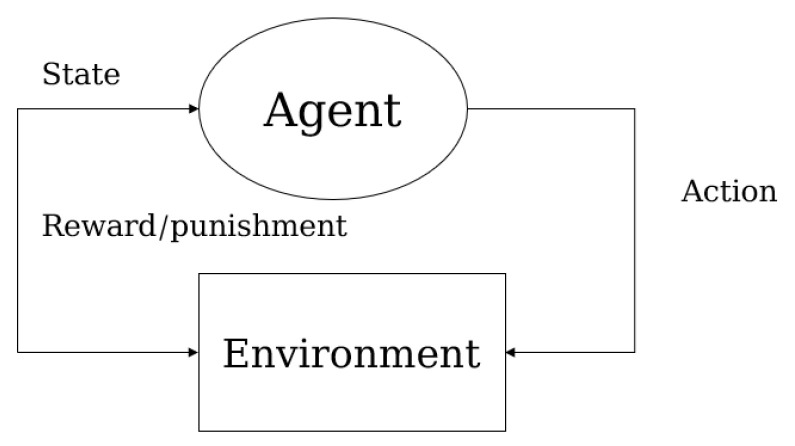
RL’s reward-driven policy optimization mechanism.

**Figure 7 sensors-25-02361-f007:**
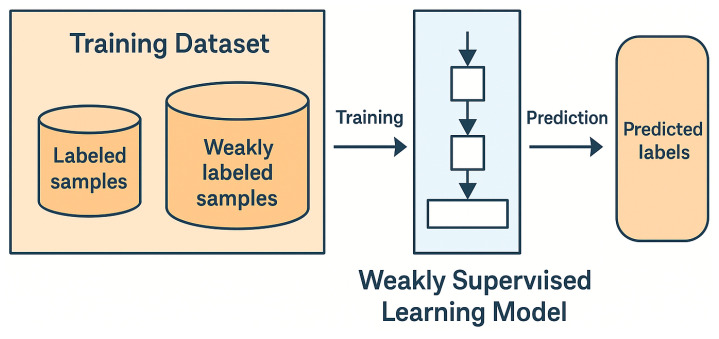
Common weakly supervised learning frameworks.

**Figure 8 sensors-25-02361-f008:**
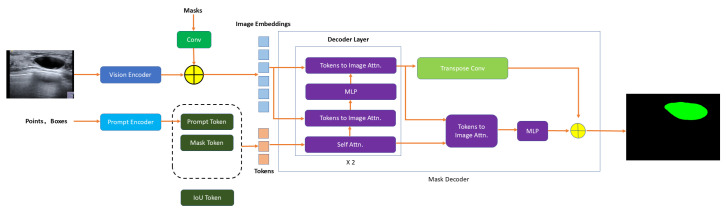
SAM’s regular network framework.

**Figure 9 sensors-25-02361-f009:**
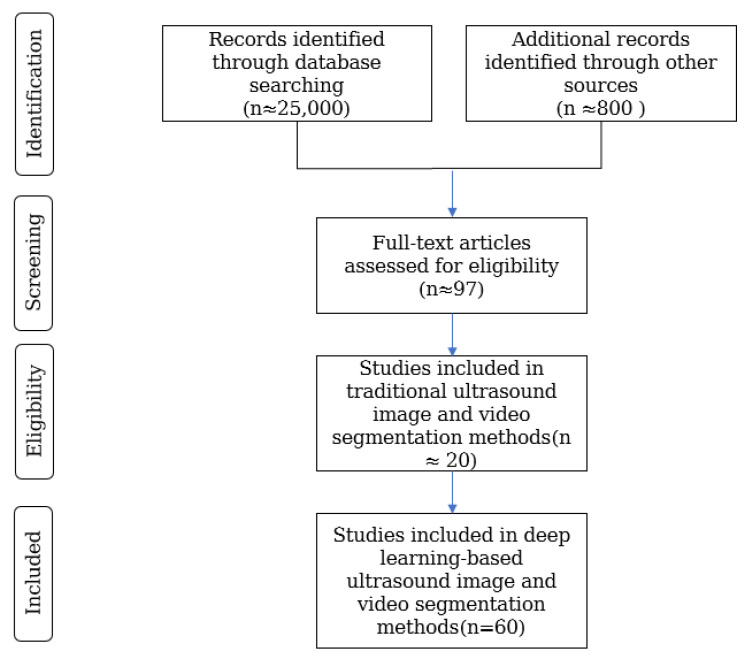
Study selection using PRISMA flowcharts.

**Table 1 sensors-25-02361-t001:** Overview of segmentation of ultrasound images and videos with U-Net.

References	ROI	Model	Dataset	Metric
Ashkani et al. [40]	Fetal abdomen	Fast and Accurate U-Net	MFP dataset	Dice: 0.9714
AMiri et al. [41]	Breast	Fine-tuning U-Net	10,000 images	Dice: 0.8 ± 0.03
Cheng et al. [43]	Lung	TL U-Net	LUS dataset	Dice: 0.8
Amiri et al. [42]	Breast	Two-stage U-Net	163 B-mode US images	Dice: 0.805
Chen et al. [44]	Breast	NU-net	BUSI and STU datasets	Dice: 0.9405 ± 0.0066
Xu et al. [45]	Breast and thyroid	MEF-UNet	BUSI, DDTL and BUS	Dice: 0.7276
Weng et al. [46]	Prostate	NAS-Unet	Promise12	Dice: 0.9737
Li et al. [47]	Transvaginal	CR-Unet	3024 TVUS	Dice: 0.912
Wang et al. [48]	Thyroid	MSAC-Unet	TND-PUH3	Dice: 0.822
Inan et al. [49]	Thyroid nodule	ResUNet++	Data obtained from Acıbadem Hospital, Istanbul, Turkey	Dice: 0.924

**Table 2 sensors-25-02361-t002:** Overview of segmentation of ultrasound images and videos with FCN.

References	ROI	Model	Dataset	Metric
Zhang et al. [50]	Lymph nodes	CFS-FCN	80 ultrasound images	F1 score: 0.858
Villa et al. [51]	Bone	FCN	1738 US images	Recall: 0.87
Xing et al. [52]	Breast	SPCGAN	670 ultrasound images	Dice: 0.93
Hu et al. [53]	Breast	DFCN + PBAC	570 BUS images	Dice: 0.8897
Li et al. [54]	Fetal head	FCN	COCO dataset	Dice: 96.96 ± 2.43%
Qian et al. [55]	Breast	FCN	Breast ultrasound dataset	Dice: 0.9224
Feng et al. [56]	Prostate	Multi-stage FCN	CCH-TRUSPS	Dice: 0.9490
Xu et al. [57]	Uterus	LGRNet	100 videos, each with 50 frames	Dice: 0.775

**Table 3 sensors-25-02361-t003:** Overview of segmentation of ultrasound images and videos with LSTM and GRUs.

References	ROI	Model	Dataset	Metric
Pan et al. [58]	Breast	Bidirectional LSTM	ABUS image dataset	Dice: 0.8178
Noort et al. [59]	Levator ani muscle	CLSTM U-net	Acquired by [66]	Dice: 0.7
Yang et al. [60]	Prostate	RNN with shape priors	300 prostate ultrasound images	DSC: 0.9239
Webb et al. [61]	Thyroid	DeepLabv3 + CLSTM	198 cineclips	Recall: 0.90
Horng et al. [62]	Median nerve	U-Net+ LSTM	420 frames of images	Dice: 0.8975
Anas et al. [64]	Prostate	CNN + GRU	29,394 images	Dice: 0.93
Devisri et al. [63]	Fetal	LSTM + U-Net	HC18	Not provided
Khdhir et al. [65]	Breast	GLCM + GRU	Wisconsin Diagnostic Breast Cancer	Dice: 0.69

**Table 4 sensors-25-02361-t004:** Overview of generating adversarial networks and diffusion models in ultrasound image and video segmentation.

References	ROI	Model	Metric
Liang et al. [67]	Thyroid	Autoencoder-GAN + VAE	Dice: 0.5111
Liang et al. [68]	Lung, hip joint, and ovary	GAN + U-Net	Dice: 0.8876
Bargsten et al. [69]	Artery layer and lumen	SpeckleGAN + U-Net	Dice: 86.02 ± 0.44%
Fatima et al. [70]	Cardiology	PatchGAN + U-Net	Dice: 0.961
Tang et al. [71]	Breast	MGCC + LDM	F1 score: 62.53 ± 2.99%
Stojanovski et al. [72]	Heart	Denoising diffusion model	Dice: 88.6 ± 4.91%
Katakis et al. [73]	Musculoskeletal	Diffusion model	Dice: 0.80
Stevens et al. [74]	Heart	Joint posterior sampling Framework	Not provided
Yao et al. [75]	Breast	DFCG	Dice: 0.7920

**Table 5 sensors-25-02361-t005:** Overview of weakly supervised deep learning methods and reinforcement learning for ultrasound image and video segmentation.

References	ROI	Model	Dataset	Metric
Girum et al. [80]	Colorectal	RL + CNN	Acquired by [83]	Dice: 0.97
Li et al. [81]	Breast	Weakly supervised learning	1389 BUS images	Dice: 83.0 ± 11.8%
Li et al. [82]	Breast	CNN + spatial pyramid module	2805 BUS images	Dice: 73.5 ± 18.0%
Sahba et al. [76]	Prostate	RL agent for sub-images	30 TRUS images	Area overlap: 90.65
Mathews et al. [78]	Lung	Unsupervised RL framework	LUS dataset	F1 score: over 44 ± 1.7%

**Table 6 sensors-25-02361-t006:** Datasets of ultrasound images and videos.

Dataset	ROI	Number	URL
Breast Ultrasound Images Dataset	Breast	780	https://www.kaggle.com/datasets/aryashah2k/breast-ultrasound-images-dataset, accessed on 3 January 2025.
DDTI: Thyroid Ultrasound Images	Thyroid	134	https://www.kaggle.com/datasets/dasmehdixtr/ddti-thyroid-ultrasound-images, accessed on 3 January 2025.
US Simulation and Segmentation	Abdomen	Not provided	https://www.kaggle.com/datasets/ignaciorlando/ussimandsegm, accessed on 3 January 2025.
CAMUS-Human Heart Data	Heart	Not provided	https://www.kaggle.com/datasets/shoybhasan/camus-human-heart-data, accessed on 3 January 2025.
Fetal Health Classification	Fetal	2126	https://www.kaggle.com/datasets/andrewmvd/fetal-health-classification, accessed on 3 January 2025.
A New Dataset and A Baseline Model for Breast Lesion Detection in Ultrasound Videos	Breast	404 frames	https://github.com/jhl-Det/CVA-Net, accessed on 3 January 2025.
Mus-V: Multimodal Ultrasound Vascular Segmentation	Neck	105 videos	https://www.kaggle.com/datasets/among22/multimodal-ultrasound-vascular-segmentation, accessed on 3 January 2025.

**Table 7 sensors-25-02361-t007:** Definition of the indexes.

Category	Positive	Negative
Actual Positive	True Positive (TP)	True Negative (TN)
Actual Negative	False Positive (FP)	False Negative (FN)

**Table 8 sensors-25-02361-t008:** Quantitative evaluation results on BUSI dataset. ↑ means that higher is better and ↓ means that lower is better. The optimal outcomes are denoted using bold typeface.

Method	Acc (%) ↑	Se (%) ↑	Dice (%) ↑	IoU (%) ↑	HD (mm) ↓
U-Net (Ronneberger et al., 2015 [22])	97.67	86.18	80.38	70.80	6.68
SegNet (Badrinarayanan et al., 2017 [91])	98.93	87.34	82.90	72.71	6.73
DeepLabV3+ (Chen et al., 2018 [92])	96.52	84.51	79.25	68.90	7.17
U-Net++ (Zhou et al., 2018 [93])	98.19	87.60	83.56	73.36	6.43
PraNet (Fan et al., 2020 [94])	98.90	86.82	83.00	72.83	6.85
RF-Net (Wang et al., 2021 [95])	98.14	86.63	83.27	73.09	6.68
TransResUnet (Tomar et al., 2022 [96])	98.95	87.35	83.74	74.62	6.38
SAMUS (Lin et al., 2023 [97])	99.29	88.52	85.89	76.36	6.16
BUSSAM (Tu et al., 2024 [84])	**99.32**	**89.16**	**86.59**	**77.21**	**6.14**

## Data Availability

No new data were created or analyzed in this study. Data sharing is not applicable to this article.

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
