# Peer review of "Deep Learning-Based Medical Ultrasound Image and Video Segmentation Methods: Overview, Frontiers, and Challenges"

_sensors, 2025, doi:10.3390/s25082361_

Round 1
Reviewer 1 Report
Comments and Suggestions for Authors
The author reviews the ultrasound image and video segmentation methods based on deep learning technology, summarizes the latest progress in this field, such as diffusion models and segmentation models, and summarizes the classical methods into several different technical paths, and presents the data set and results to help readers quickly understand the field. This review is very meaningful. However, some content needs to be improved.
- Figure3, the images can be colorful.
- The author categorizes different methods according to different technical paths such as U-Net and its variants, fully convolutional neural networks, recurrent neural networks, etc. These classifications are not appropriate, and there are intersections or belonging relationships between them. For example, the unet method (fully convolutional neural networks) and its category can be classified as fully convolutional neural networks. However, weakly supervised learning and this classification are not parallel. Weakly supervised learning is divided according to the way of learning labels, and they are not parallel. The author can consider the classification again to arrange the structure of the paper.
- As a review paper, when introducing different technical paths, it is better to briefly introduce the relevant technical details through images or mathematical expressions so that readers can better understand rather than simply listing the literature. For example, the author can introduce the structure of RNN, weak supervision methods, etc. in detail.
Author Response
Point 1: Figure3, the images can be colorful.
Response 1: We sincerely appreciate the reviewers' valuable comments and suggestions, which have significantly improved the quality of our manuscript. We have carefully considered these suggestions and revised the paper accordingly. Once again, we are deeply grateful for the reviewers' time and effort. Since the text of the picture is not clear, the structure diagram presented in Figure 3 is displayed within the corresponding chapter (Figure 3/4/5/6) and has been updated with color enhancements and clearer text for improved readability.
Point 2: The author categorizes different methods according to different technical paths such as U-Net and its variants, fully convolutional neural networks, recurrent neural networks, etc. These classifications are not appropriate, and there are intersections or belonging relationships between them. For example, the unet method (fully convolutional neural networks) and its category can be classified as fully convolutional neural networks. However, weakly supervised learning and this classification are not parallel. Weakly supervised learning is divided according to the way of learning labels, and they are not parallel. The author can consider the classification again to arrange the structure of the paper.
Response 2: In Section 2, the original technical path classification is changed to four categories according to the learning paradigm: supervised learning, unsupervised learning, weakly supervised learning, and deep reinforcement learning for more parallel classification, and the previous categories are reclassified into two levels to describe related work. Supervised learning includes FCNs,Unet and its variants, and RNNs. Unsupervised learning includes GAN and Diffusion Model. Weakly supervised learning includes common weakly supervised learning methods and SAM (which is classified as weakly supervised learning due to full supervision and training of large models and special interactions), and deep reinforcement learning.
Point 3: As a review paper, when introducing different technical paths, it is better to briefly introduce the relevant technical details through images or mathematical expressions so that readers can better understand rather than simply listing the literature. For example, the author can introduce the structure of RNN, weak supervision methods, etc. in detail.
Response 3: In the part of introducing the technical path, the corresponding formula introduction is added. Lines 161 to 176 describe the working principle of LSTM and GRU in RNN in more detail with the formula, and lines 284 to 300 add the formula to introduce the working principle of general weakly supervised learning in more detail. In addition, we added Figure 7 and Figure 8 to illustrate the structure of weakly supervised learning and SAM respectively.

Reviewer 2 Report
Comments and Suggestions for Authors
The authors provide an overview about ultrasound image and video segmentation methods that are using deep learning techniques. Generally, the paper is well presented and easy to follow. The categorization and assessment of state-of-the-art techniques is appreciated. However, the paper suffers from the following:
1) The abstract and conclusion are not supported by results or the importance of integrating deep learning into ultrasound image and video segmentation.
2) The title of Figure 1 should be more concise.
3) The selection of papers included in the survey is not clear. What inclusion and exclusion criteria are used in the selection? It is obvious from section 3 that the authors have used a set of keyboards to find papers, however the selection process is not mentioned.
4) The difference between this survey and other existing one is not existing.
5) Text inside most of figures is not visible.
6) In section 2, the authors have: i) change the title since it is too generic, and 2) provide critical analysis of exiting solutions and not only describe them.
7) Please provide clear text before the first use of any abbreviations, for example PRISMA, etc.
8) In Tables 1-6, the authors perform a comparative analysis of exiting techniques according to different datasets used in each one. Unfortunately, this evaluation is not fair unless a comparative with same dataset and same environment are fixed.
9) Section 4 is not useful until a simulation is performed.
Overall, the paper is just mentioning and categorization existing papers in a specific domain while it misses totally any theoretical or experimental aspects. Accordingly, I advise to discuss the following points to enhance the quality of the paper before any possible publication:
10) Discuss the existing market solutions of medical ultrasound image and video segmentation.
11) Add a section to separately discuss the challenges facing by applying deep learning in medical segmentation.
12) Add a section to discuss the real-life applications and projects of deep learning into medical segmentation.
13) Select a list of existing solutions then to perform a comparative study in a fair simulation environment.
Author Response
Point 1: The abstract and conclusion are not supported by results or the importance of integrating deep learning into ultrasound image and video segmentation.
Response 1: A description of the results is added in lines 845 through 866 of the conclusion section. The changes can be tracked in the revision (RevisionTracked.pdf), which are marked up by orange marker “According to Comment.1 of R2”.
Point 2: The title of Figure 1 should be more concise.
Response 2: The original title was changed to the current "Deep learning is a branch of artificial intelligence and machine learning. Deep learning is a branch of artificial intelligence and machine. According to the classification of learning paradigm, it is divided into more detailed secondary classification." In addition, the content structure of Figure 1 has been reorganized more precisely.
Point 3: The selection of papers included in the survey is not clear. What inclusion and exclusion criteria are used in the selection? It is obvious from section 3 that the authors have used a set of keyboards to find papers, however the selection process is not mentioned.
Response 3: In lines 312 to 334 of the revised manuscript, the selection criteria of literature were revised more clearly, including how to select and exclude the references.
Point 4: The difference between this survey and other existing one is not existing.
Response 4: Thanks a lot for your constructive comments on our work. Our manuscript focuses on deep learning segmentation of ultrasonic images and videos at the same time. According to our investigation, previous survey papers tend to focus primarily on ultrasonic images. In addition, our survey is more up-to-date, including the latest large model techniques (such as SAM). Most of all, the practical application level analysis is also added up according to your valuable comments, which makes our survey more complete and unique.
Point 5: Text inside most of figures is not visible.
Response 5: The text of all illustrations has been optimized and the text of figures can now be read more clearly.
Point 6: In section 2, the authors have: i) change the title since it is too generic, and 2) provide critical analysis of exiting solutions and not only describe them.
Response 6: In Section 2, we have optimized titles and mainly propose an introduction to the working principles of various deep learning approaches. In the following Section 3, we make a critical analysis of the corresponding approaches to ultrasonic image and video segmentation.
Point 7: Please provide clear text before the first use of any abbreviations, for example PRISMA, etc.
Response 7: Thanks very much for your careful work. All abbreviations have been clearly described when first mentioned, including PRISMA, etc.
Point 8-9: In Tables 1-6, the authors perform a comparative analysis of exiting techniques according to different datasets used in each one. Unfortunately, this evaluation is not fair unless a comparative with same dataset and same environment are fixed. Section 4 is not useful until a simulation is performed.
Response 8-9: Thanks for the reviewer's comments. Because most of the experimental data and source codes in the literature are not public, we considered that the table header ‘Evaluation’ in Table 1-6 might cause misunderstanding, so we changed the table header to ‘Metric’. We have revised the introduction of section 4 (lines 699-703), which mainly aims to introduce indicators used in the respective literature for evaluating experimental results. At the same time, we have supplemented the comparison of different approaches under the same public dataset in Table 8, which is a fair simulation result.
Point 10: Discuss the existing market solutions of medical ultrasound image and video segmentation.
Response 10: According to the reviewer's comments, we have a separate subsection 5.2 describing the solutions on the market. In lines 802-832, we summarize the industry reports, white papers, etc. of the major medical device companies in the market, summarizing the practical application projects and solutions of ultrasound image and video segmentation in the market.
Point 11: Add a section to separately discuss the challenges facing by applying deep learning in medical segmentation.
Response 11: The subsection 5.3 entitled ‘Challenges and opportunities’, discussing challenges in the application of ultrasound image and video segmentation based on current deep learning is added in lines 833 through 844.
Point 12: Add a section to discuss the real-life applications and projects of deep learning into medical segmentation.
Response 12: According to the reviewer's comments, in a new Section 5.2 from lines 802 to 832, we discuss the practical application of deep learning in medical segmentation by summarizing the landing projects of major medical device companies, including Philip, GE, etc., as well as the products of emerging technology companies, and some medical image segmentation competitions.
Point 13: Select a list of existing solutions then to perform a comparative study in a fair simulation environment.
Response 13: Thank you for your careful review and valuable suggestions. We have supplemented the comparison of different approaches under the same public dataset in Table 8, which is a fair simulation result. The corresponding text is described in the first paragraph of Section 4.1.

Round 2
Reviewer 1 Report
Comments and Suggestions for Authors
It has met the requirements.
Reviewer 2 Report
Comments and Suggestions for Authors
The authors addressed all my issues and concerns. I suggest the publication of the paper in the journal.